

# RUNX1 facilitates heart failure progression through regulating TGF-β-induced cardiac remodeling

Peng Qi[1], Qian Zhai[1] and Xiquan Zhang[2]

[1] Department of Cardiac Surgery Intensive Care Unit, Qilu Hospital of Shandong University, Jinan, China
[2] Department of Cardiac Surgery, Qilu Hospital of Shandong University, Jinan, China

## ABSTRACT

**Background:** Heart failure is caused by acute or chronic cardiovascular diseases with limited treatments and unclear pathogenesis. Therefore, it is urgent to explore new therapeutic targets and reveal new pathogenesis for heart failure.

**Methods:** We carried out heart failure animal model by transverse aortic arch constriction (TAC) in mice. The left ventricular internal diameter diastole (LVIDd), left ventricular internal diameter systole (LVIDs), and ejection fraction (EF) value were detected using ultrasound and myocardial fibrosis was evaluated by Masson stain assay. Cell apoptosis in myocardial tissues were detected by TUNEL immunofluorescence stain. Signal pathway analysis was performed by dual-luciferase reporter assay and western blot.

**Results:** Our results showed that inhibition of RUNX1 led to remission of cardiac enlargement induced by TAC in mice. Inhibition of RUNX1 also caused raise of EF and FS value under TAC-induced condition. Besides, RUNX1 inhibition mice showed decreased myocardial fibrosis area under TAC-induced condition. RUNX1 inhibition caused decrease of apoptotic cell rate in myocardial tissues under TAC. Interestingly, we found that RUNX1 could promote the activation of TGF-β/Smads in dual-luciferase reporter assay.

**Interpretation:** We illustrated that RUNX1 could be considered as a new regulator of myocardial remodeling by activating TGF-β/Smads signaling. Based on this, we concluded that RUNX1 may be developed as a new therapeutic target against heart failure in the future. In addition, this study also provide a new insight for the etiological study on heart failure.

## INTRODUCTION

Heart failure is a group of clinical syndromes caused by kinds of heart diseases such as coronary heart disease, congenital heart disease, dilated cardiomyopathy et al (*Costa, 2022*; *Delgado et al., 2022*; *Dhalla, Bhullar & Shah, 2022*). Heart failure can be considered as the end stage of various heart diseases. The causes of heart failure are still unclear, but scholars worldwide have identified that cardiac remodeling is the main cause and key step of heart failure progression (*Costa, 2022*; *Delgado et al., 2022*; *Dhalla, Bhullar & Shah, 2022*).

Corresponding authors
Qian Zhai, 13793182152@163.com
Xiquan Zhang, zhangxiquan196411@163.com

It is known that the inhibition of cardiac remodeling process can significantly improve cardiac function and prognosis of heart failure patients (*Bai et al., 2022*; *Bevere et al., 2022*; *Cavallo et al., 2022*). During clinical treatment, clinicians not only focus on improvement of hemodynamics but also aim to delay cardiac remodeling by using angiotensin receptor blockers (ARBs) (*Chen et al., 2022*; *Choy et al., 2022*; *Figueiredo et al., 2022*; *Gu et al., 2022*). However, causes of cardiac remodeling are complex and clinical treatment using ARBs only cause improvement of cardiac remodeling in some of the heart failure patients. The usage of β-receptor blocker and angiotensin converting enzyme inhibitor (ACEI) can lead to decrease of hospitalization in heart failure patients but no significant decline of morbidity and mortality of heart failure (*Liu et al., 2022*; *Lv et al., 2022*; *Pan et al., 2022*). Thus, it is urgent to explore new mechanisms of cardiac remodeling and find new molecular target against heart failure.

The salient features of cardiac remodeling can be summarized as myocardial structure change, myocardial cell hypertrophy with mitochondria decrease, increase of myocardial cell apoptosis (*Soulat-Dufour et al., 2022*). The stromal elements changes of cardiac remodeling include excessive collagen accumulation and imbalance of extracellular matrix synthesis and degradation (*Stassen et al., 2022*). It is important to find new approach or target drugs for preventing myocardial cell apoptosis and excessive myocardial fibrosis.

Myocardial cell apoptosis and myocardial fibrosis can be regulated by many signal pathways such as TGF-β/Smads signaling, P38/MAPK signaling, RhoA/ROCK signaling, Wnt signaling and so on (*Adel et al., 2022*; *Banks et al., 2022*; *Chen et al., 2022*). Among kinds of cardiac remodeling related signaling pathways, TGF-β/Smads pathway can be considered as the most important regulator in excessive extracellular matrix (ECM) accumulation process of cardiac remodeling (*Han et al., 2022*; *Iyer et al., 2022*; *Ji et al., 2022*; *Li et al., 2022*; *Liang et al., 2022*). TGF-β related signal molecule widely participate in regulating cardiac fibroblasts proliferation, differentiation, migration and ECM production (*Chen et al., 2022*). The TGF-β1 can obviously promote synthesis of collagen type I and type III by cardiac fibroblasts and inhibit the degradation of them. In a word, TGF-β signaling pathway play an important role in promoting myocardial fibrosis process. Therefore, proteins or cytokines which regulate TGF-β signaling activity can be chosen as potential target molecule against myocardial fibrosis.

Runt-related transcription factor1 (RUNX1) belongs to Runx family which includes RUNX1, RUNX2, RUNX3. Previous studies have shown that RUNX1 is critical for generating definitive hematopoietic stem cells through endothelial-to-hematopoietic transition (EHT) (*Almazni et al., 2022*; *Ariffin, 2022*; *Bang et al., 2022*; *Bao & Guo, 2022*; *Chang et al., 2022*). It was reported that RUNX1 can transfer from plasma to nucleus and regulate target gene expression. Moreover, RUNX1 participated in collagen construction through interact with extracellular matrix in gastric cancer. Besides, it is also reported that RUNX1 plays an important role in regulating renal fibrosis by promoting TGF-β induced renal tubular epithelial-to-mesenchymal transition (EMT) (*Jakobczyk et al., 2022*; *Jeong et al., 2022*; *Li & Jia, 2022*; *Lin, 2022*). Interestingly, Runx family of transcription factors such as RUNX1 are known as co-activators to interact with Smads and play an important

role in regulating TGF-β/Smads signaling. That is why we pay so much attention to explore the regulatory role of RUNX1 in TGF-β induced cardiac remodeling.

We reviewed many literatures about RUNX1 in different diseases and deeply explored the differential expression of this gene in heart diseases. The GEO dataset GSE226314 revealed that RUNX1 as a possible therapeutic target to facilitate cardiac recovery (*Amrute et al., 2023*). These findings encouraged us to choose RUNX1 as potential heart failure treatment target. Until now, the function of RUNX1 in regulating myocardial fibrosis and cardiac remodeling remains unclear. Therefore, we chose RUNX1 as target in this study.

In this study, we used a classical heart failure mouse model that performed by transverse aortic arch constriction (TAC) and explored whether and how RUNX1 regulated cardiac remodeling. Our results showed that RUNX1 promoted myocardial fibrosis and cardiomyocyte apoptosis in TAC-induced mice heart failure model. Since the mice heart failure model was established and the regulatory effect of RUNX1 on cardiac remodeling was investigated, we mainly pay attention to illustrate the mechanisms of how RUNX1 regulate cardiac remodeling *via* activating TGF-β/Smads signaling. Thereby, we carried out this study and identified RUNX1 as an important regulator in cardiac remodeling and provided a new potential molecular target for anti-heart failure therapy.

## MATERIALS AND METHODS

### Animal experiment

This study was approved by the Animal Ethics Committee of Shandong University Qilu Hospital (IACUC approval number, KYLL-2022(ZM)-741). C57BL6 mouse was chosen as heart failure model animal. All the C57BL6 mice were purchased from animal experiment center of Shandong University. The C57 mice were fed with normal chow in SPF condition. The C57 mice involved in this study were raised in quiet environment at density of six mice per cage in normal circadian rhythm. The heart failure model was carried out using transverse aortic arch constriction method (TAC).

Mice anesthesia was performed using tracheal intubation method. Tracheal intubation anesthesia was carried out by using inhalation 5% Isoflurane at speed of 0.8–1.0 L/min and maintain dose using 3% Isoflurane at speed of 0.8–1.0 L/min. The mice neck skin were cut and trachea were exposed by blunt dissection method. Tracheal intubation need to be inserted from midline of mandible until tracheal carina position. A ventilator was used continuously during tracheal intubation anesthesia and tidal volume was set at 4 ml/min with respiratory rate at 75–90 times/min.

The C57 mice in TAC model group were fixed and performed thoracotomy and aortic arch ligation under anesthesia with tracheal intubation. The C57 mice in control group were only performed thoracotomy without aortic arch ligation under anesthesia with tracheal intubation. All the surviving animals at the conclusion of this experiment were fed at SPF condition until naturally dead. Data collection was performed eight weeks after TAC experiment.

## Cell lines

The HL-1 Cardiac Muscle Cell Line was purchased from Sigma-Aldrich (No. SCC065) and was used for luciferase reporter assay. Cell line provenance was provided as a Supplemental File.

## Groups and administration

At four weeks after heart failure model was established, the mice were randomly divided into three groups, TAC+lenti-NC group, TAC+lenti-shRUNX1 group and control group. The mice in control group were treated with thoracotomy only. The mice in TAC+lenti-NC group accepted thoracotomy and aortic arch ligation and were treated with nonsense lentivirus vector by tail vein injection. The mice in TAC+lenti-shRUNX1 group were treated with TAC and lentivirus vector which carried RUNX1-shRNA by tail vein injection. In addition, RUNX1-shRNA sequences were as follows, CCTCGAAGACATCGGCAGAAA; and lenti-NC sequences were, CCTAAGGTTA AGTCGCCCTCG.

## Cardiac function and structure noninvasive assessment

We used cardiac ultrasound to detect cardiac function and left ventricle structure changes in mice under anesthesia by 3% pentobarbital sodium intraperitoneal injection.
The evaluating parameters including left ventricular internal diameter diastole (LVIDd), left ventricular internal diameter systole (LVIDs), ejection fraction (EF) value and fractional shortening of ventricle (FS) value. Both EF and FS value are important pathophysiological parameters for myocardial systolic function assessment. Besides, LVIDd combing with LVIDs and EF value can be used as parameters for systolic heart failure assessment.

## Construction of lentivirus vector

HEK-293T cells in the logarithmic phase were used for lentivirus package. We seeded 293T cells into 15 cm$^2$ culture dish at density of $5 \times 10^5$ cells/mL and cultured cells using bovine serum-free medium 2 h before cell transfection. Cell transfection was carried out using Lipofectamine 2,000 when cell growth density reached to 70–80%. Four vectors system was chosen in this lentivirus transfection assay which includes PACK vectors (pPACKH1-GAG, pPACKH1-REV, pPACKH1-VSV), small hairpin RNA targeting RUNX1 (pLVX-sh-RUNX1) or nonsense small hairpin RNA control vectors (pLVX-sh-NC). All the PACK vectors, shRNA recombinant vectors were mixed with Lipofectamine, Polybrene and bovine serum-free medium at room temperature for 20 min. The HEK-293T cells were washed by PBS (phosphate buffer solution) for three times before transfection. We added mixture of vectors and Lipofectamine into 293T cells drop by drop. After that, the 293T cells were cultured using medium containing 10% fetal bovine serum and puromycin for 48 h at 37 °C and 5% $CO_2$ condition. Subsequently, the supernatant of HEK-239T cells was collected and slightly centrifuged at 3,000×g at 4 °C for 10 min. We filtered the gained cell supernatant using 0.45 µm filter membrane and froze the virus liquid at −80°C refrigerator.

## Detection of blood BNP level

BNP (natriuretic peptide type B) is generally accepted as assessment marker for heart failure. In this study, we collected blood from ventricle of mice that were killed by cervical dislocation. A total of 0.5 ml serum was gained from mice blood and equivalent buffer was added. The specific procedures were carried out following the protocol of BNP ELISA kit (RAB0386, minimum detectable concentration is 1.66 pg/mL, detection range, is 0.1–1,000 pg/mL; Sigma-Aldrich, St. Louis, MO, USA) and absorbance value at 450 nm were detected by microplate reader.

## Pathomorphological assessment of myocardial tissue

The pathomorphological tissue changes of myocardium were evaluated by H&E staining. All the myocardial tissues were dehydrated by gradient ethanol step by step and embedded by paraffin. The embedded tissue were cut into serial histologic sections and dewaxed using xylene step by step. The tissue slides were stained by eosin dye and Hematoxylin according to H&E staining procedures. The changes of myocardial fibrosis were assessed by Masson staining. For Masson staining, tissue slides were stained by Weigert hematoxylin, Ponceau and Aniline blue step-by-step. In addition, the pale blue in Masson staining image is for collagen fibers, pink for muscle fibers and blue brown represents nuclei.

## Detection of apoptotic cells

TUNEL staining was applied to apoptotic rate detection in myocardial tissues. For TUNEL staining, we stained myocardial tissue slides with TUNEL Assay kit (BrdU-Red, ab66110; Abcam, Cambridge, UK) following the procedure instructions and the fluorescence images were captured by using Fluorescence microscope. At the same time, we stained cardiomyocyte in tissue slides with Desmin antibody (Alexa Fluor 555 Anti-Desmin antibody, ab203422; Abcam, Cambridge, UK shown in green) at 1/100 dilution by immunofluorescence staining assay. Moreover, the nuclei in tissue slides were stained by using DAPI (ab285390; Abcam, Cambridge, UK). DAPI was used for nuclei stain at concentration of 10 μg/ml under room temperature for 15 min avoid of light.

## Dual-luciferase reporter assay

HL-1 cells were plated in 96-well plates and were transiently transfected with 0.15 μg of pGL4.48[luc2P/SBE/Hygro] vector (Promega, Madison, WI, USA) and 0.015 μg of *Renilla* luciferase control reporter vector (Promega, Madison, WI, USA). The transfection assay above was carried out with FuGENE six transfection reagents (Promega, Madison, WI, USA) following the manufacturer's instructions. The SBE in pGL4.48 vector was the Smad binding element and luciferase activity was normalized to Renilla activity. After 24 h of transfection, HL-1 cells were treated with TGFβ1 (20 ng/ml, Peprotech, Rocky Hill, CT, USA) for 24 h. In the RUNX1 treated group, cells were treated with TGFβ1 (20 ng/ml) for 12 h and then co-treated with recombinant RUNX1 protein (Origene Technologies, Rockville, MD, USA) for 12 h. In the end, cells were collected and put into luciferase reporter assay.

## Protein extraction and western blot

The whole heart was gained from mice and the total protein was extracted using T-PER Tissue Protein Extraction Reagent (No.78510; Thermo Fisher Scientific, Waltham, MA, USA) following the manufacturer's instructions. Besides, protein quantitative detection was carried out with BCA protein detection Kit following the procedures provided by the manufacturer (Pierce™ BCA protein detection kit, No. 23227; Thermo Fisher Scientific, Waltham, MA, USA). The expressions of RUNX1, Smad2, Smad3, Smad4 and GAPDH were detected by western blot assay. The concentration of separation gel (15%) and SDS-PAGE electrophoresis voltage were adjusted according to molecular weight of target proteins. Furthermore, goat anti-Mouse IgG H&L Adsorbed Secondary Antibody (Alexa Fluor™ Plus 680, A32729; Invitrogen, Waltham, MA, USA), and goat anti-Rabbit IgG H&L Highly Cross-Adsorbed Secondary Antibody (Alexa Fluor™ Plus 680, A32734; Invitrogen, Waltham, MA, USA) were used for fluorescence imaging. All the fluorescence images were captured by using Odyssey LI-COR infrared imaging system. The primary RUNX1 antibody was purchased from Abcam (Rabbit to RUNX1, ab229482, Reactivity: Mouse, Rat, Human) and Smads antibodies from CST (Smad2 Rabbit mAb (dilution ratio, 1:2,000; #5339S), Smad3 Rabbit mAb (dilution ratio, 1:2,000; #9523), Smad4 Rabbit mAb (dilution ratio, 1:2,000;#46535), Phospho-Smad2 (dilution ratio, 1:2,000; Ser465/Ser467, Rabbit mAb#18338), Phospho-Smad3 (dilution ratio, 1:2,000; Ser423/Ser425, Rabbit mAb#9520), GAPDH (Rabbit mAb#5174, dilution ratio, 1:2,000; Cell Signaling Technology; Danvers, MA, USA)). All the fluorescence images were converted into monochrome images and grey value semi-quantitative analysis was carried out using ImageJ software.

## RNA extraction and qPCR

RNA extraction from myocardial tissues was carried out using Trizol Reagent (Invitrogen, Waltham, MA, USA) following manufacturer's instructions. All the PCR primers were designed and synthesized by Shanghai Sangon Co. (Shanghai, China). Primers for RUNX1 gene were, 5′-3′ CAGGCAGGACGAATCACACT; 3′-5′ CTCGTG CTGGCATCTCTCAT. Primers for GAPDH were designed as 5′-3′ GGTGAAG GTCGGTGTGAACG; 3′-5′ CTCGCTCCTGGAAGATGGTG. cDNA synthesis was performed using RT Revert Aid First Strand cDNA Synthesis Kit (Thermo Fisher Scientific, Waltham, MA, USA). PCR reaction was carried out by using Platinum SYBR Green PCR Super Mix- UDG w/ROX Kit (No. C11744100; Invitrogen, Waltham, MA, USA) according to the manufacturer's instructions. Fluorescence signals were captured by ABI7500 real time PCR instrument and cycles to threshold (CT value) were calculated and recorded. $2^{-\Delta\Delta CT}$ value was used to reflect relative gene expression levels.

## Statistical analysis

Data analysis was carried out using SPSS 20.0 software (Chicago, IL, USA). Student's t-test was used for comparison between two groups of numerical variable data. Besides, One-way ANOVA method was used for comparison among three groups and more numerical
variable data. $P < 0.05$ was considered as the difference with statistical significance. $P < 0.01$ was considered as the difference with obvious statistical significance.

## RESULTS

### Inhibition of RUNX1 lead to morphological changes of heart in TAC mouse

We carried out heart failure animal model by TAC induced method and found that heart size in TAC-induced group (TAC+lenti-NC) was much bigger than that in control group (Fig. 1A). Compared with lenti-NC group, mouse heart in lenti-shRUNX1 group became much smaller in size (Fig. 1A). These results revealed that the inhibition of RUNX1 caused alleviation of cardiac enlargement induced by TAC.

### Inhibition of RUNX1 lead to improvement of heart function in TAC mouse

We performed noninvasive ultrasound examination to assess cardiac function changes in different groups (Fig. 1B) and found that LVIDs and LVIDd values in lenti-shRUNX1 group were much lower than those in TAC+lenti-NC group (Fig. 1C, i, ii). Besides, EF and FS values which can reflect cardiac systolic function in lenti-shRUNX1 group mice were much higher than those in TAC+lenti-NC group (Fig. 1C, iii, iv). These results showed that the inhibition of RUNX1 can lead to improvement of cardiac function in TAC mouse. This result also means that heart function can be improved by RUNX1 inhibition in TAC induced model.

### Inhibition of RUNX1 can ameliorate myocardial fibrosis after TAC

We compared histological changes of mouse heart in different groups through H&E staining. The results showed that myocardial cells in control group were arranged with clear and regular transverse striations with normal cardiomyocyte morphology (Figs. 2A and 2B). Nevertheless, myocardial cells in TAC+lenti-NC group had irregular shape and significant disordered arrangement compared with TAC+lenti-shRUNX1 group and control group. In addition, the transverse striations of heart tissues were broken in TAC +lenti-NC group. The tissue structure and cell morphological changes in TAC+lenti-shRUNX1 group were much slighter than that in TAC+lenti-NC group.

We assessed myocardial fibrosis of mouse heart tissues in different groups by Masson's trichrome staining and found that myocardial tissue in control group showed no obvious collagen accumulation. However, myocardial tissue in TAC+lenti-NC group appeared significant collagen accumulation and a decrease of myocardial cells number compared with TAC+lenti-shRUNX1 group (Figs. 3A and 3B). Therefore, these results revealed that the inhibition of RUNX1 could lead to significant alleviation of myocardial fibrosis and cardiac remodeling in TAC mouse.

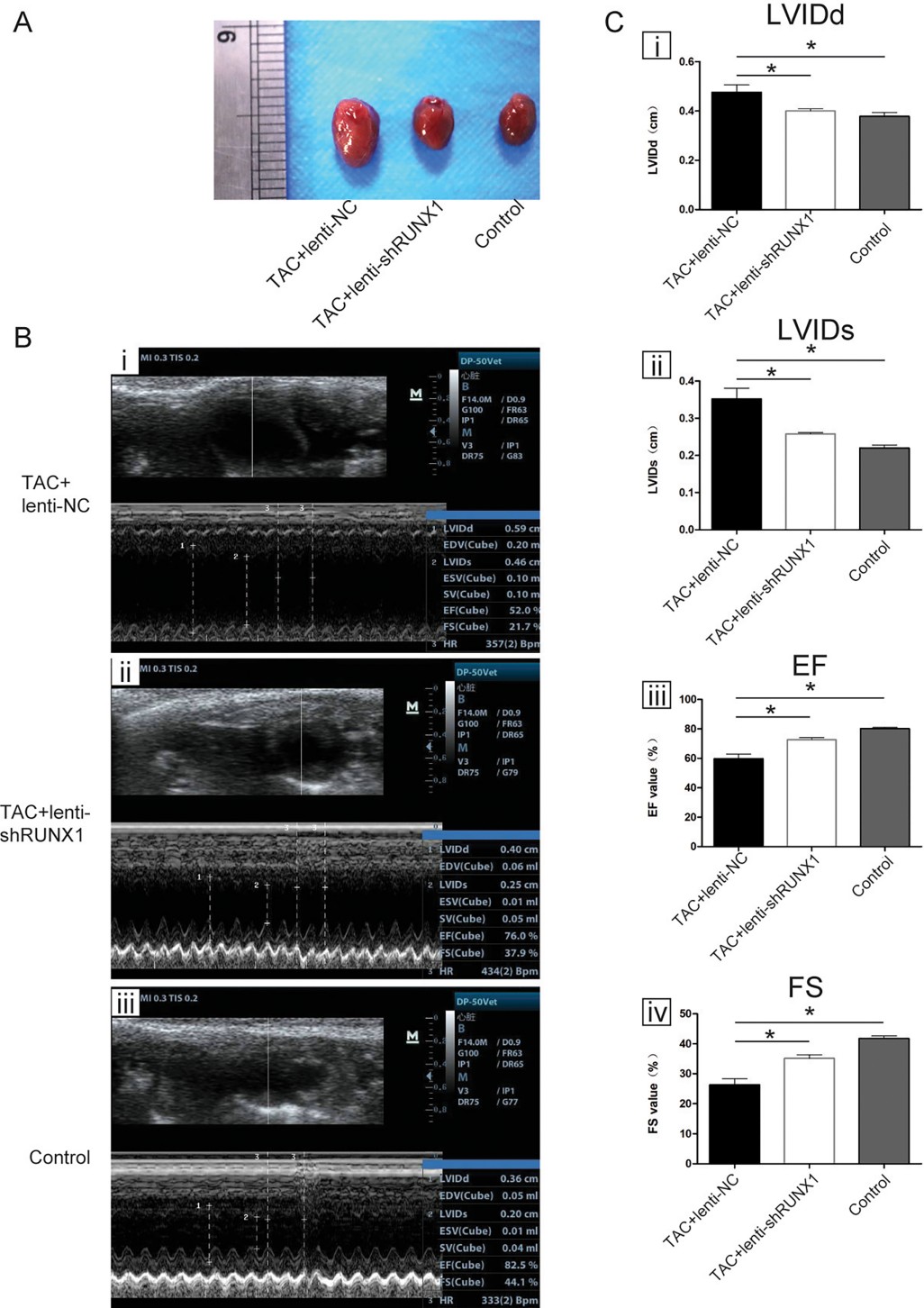

**Figure 1 RUNX1 contributes to heart failure *in vivo*.** (A) Morphological changes of mice heart in different groups under TAC-induced; (B) Ultrasound images of heart structure in different groups under TAC-induced mice; (C) Comparisons of LVIDd, LVIDs, EF and FS value among different groups. Asterisks (*) indicate a significant difference.

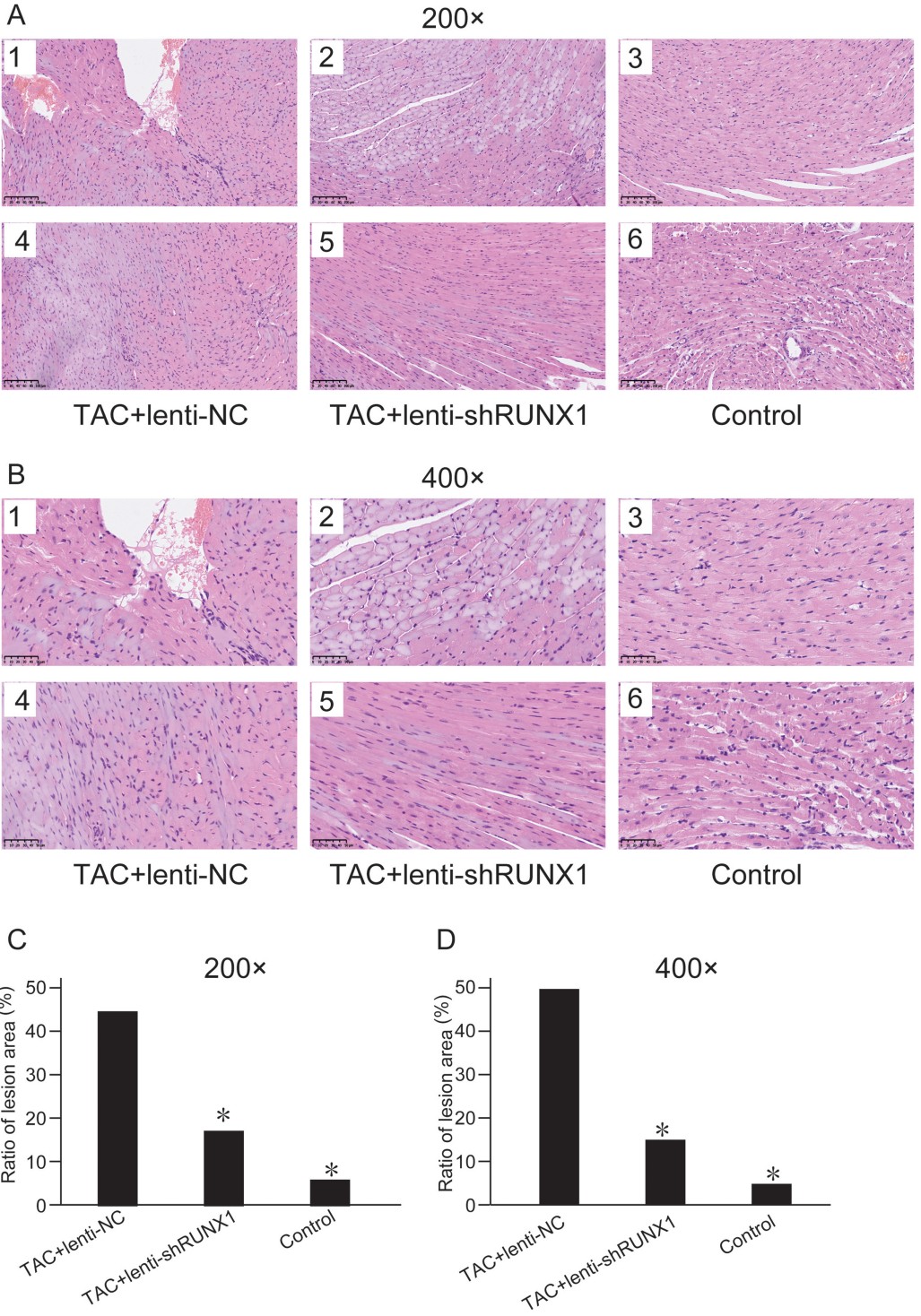

**Figure 2 HE staining images of heart tissues.** HE staining images of heart tissues in control, TAC +lenti-NC, TAC+lenti-shRUNX1 groups under 200× (A) and 400× (B) amplification. Asterisks (*) indicate a significant difference.               

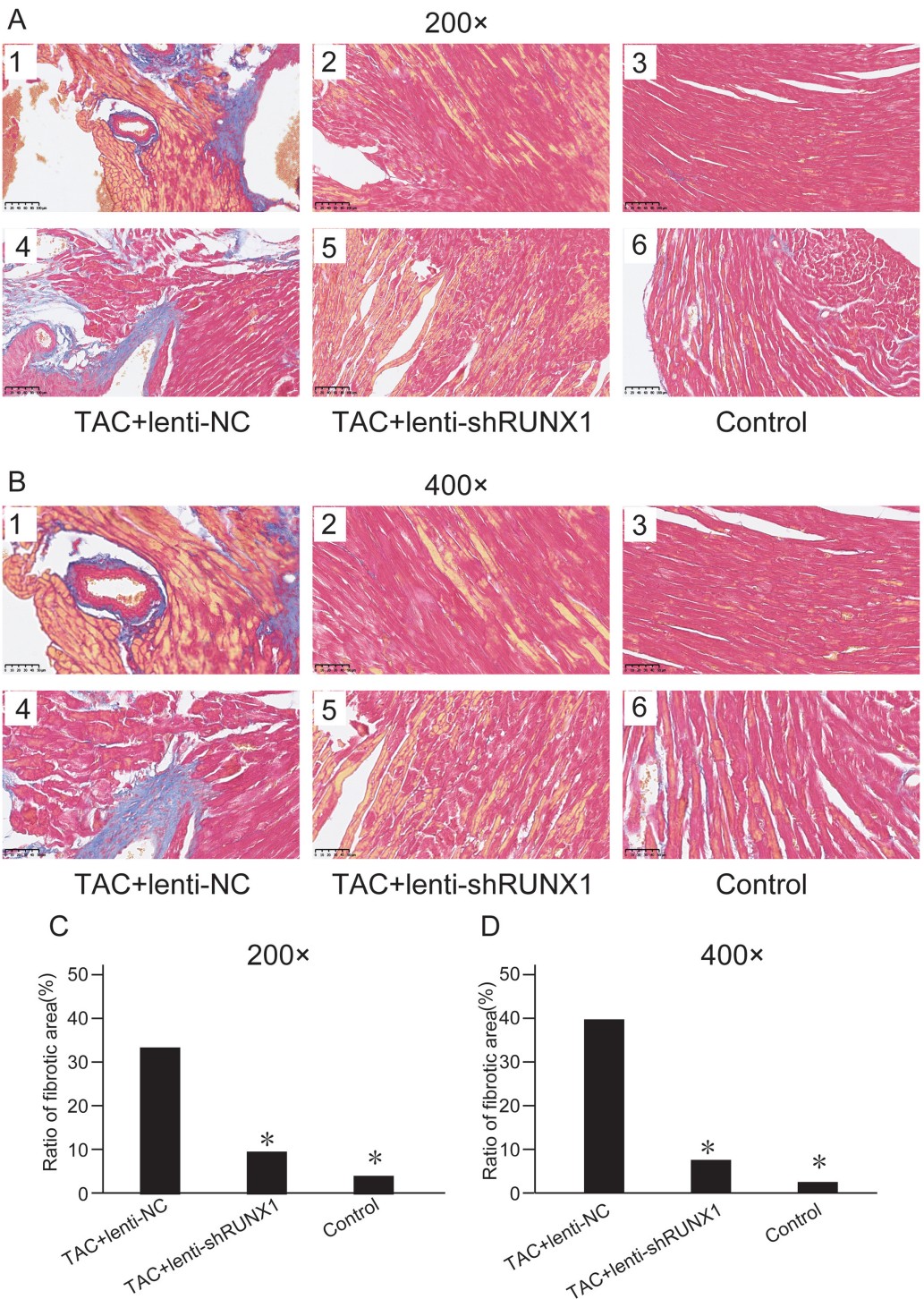

**Figure 3  Masson's trichrome staining images.** Masson's trichrome staining images of heart tissues in control, TAC+lenti-NC, TAC+lenti-shRUNX1 groups under 200× (A) and 400× (B) amplification. Asterisks (*) indicate a significant difference.

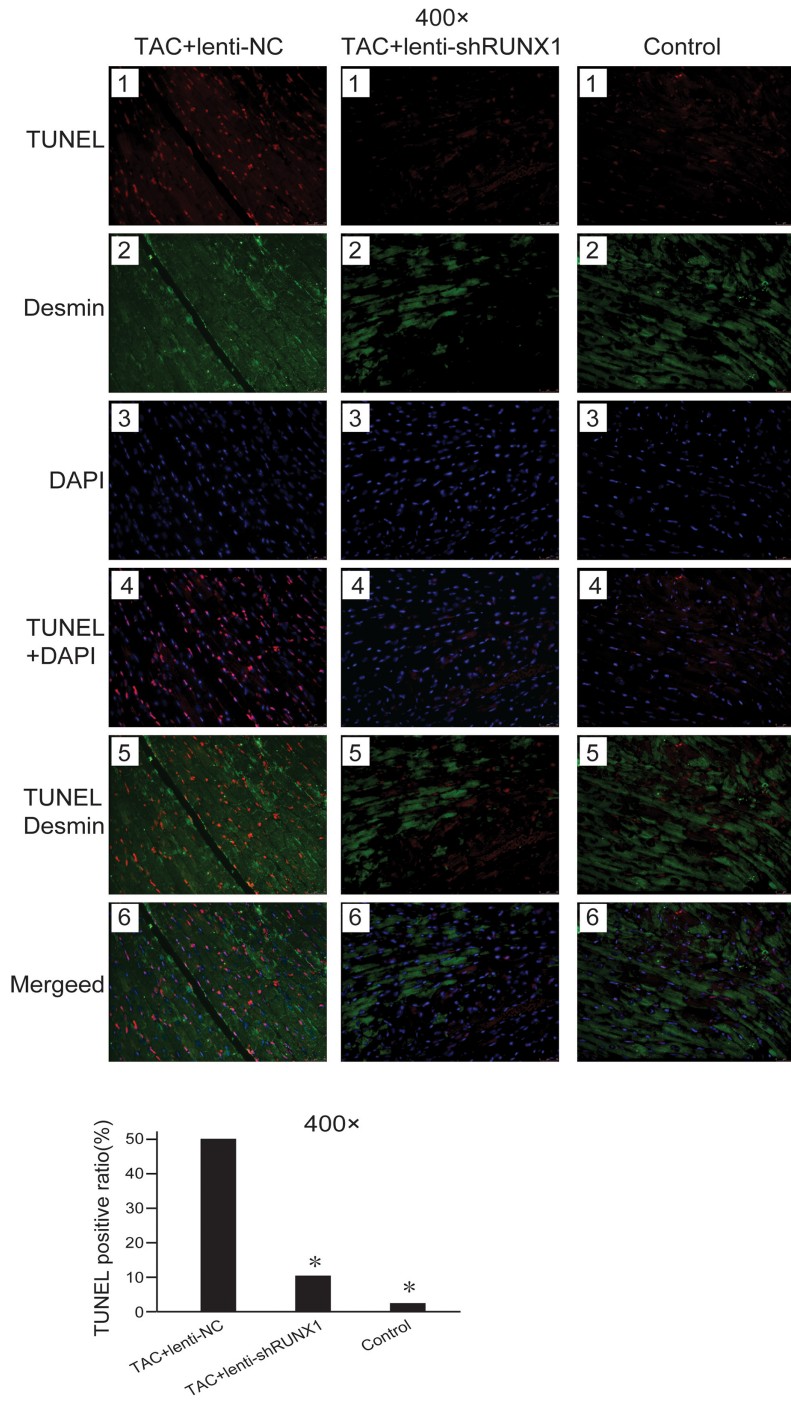

**Figure 4 Fluorescence microscope image of heart tissues.** Fluorescence microscope image of heart tissues in control, TAC+lenti-NC, TAC+lenti-shRUNX1 groups. Red fluorescence images for TUNEL, green fluorescence images for Desmin, blue fluorescence images for DAPI. Asterisks (*) indicate a significant difference.                              

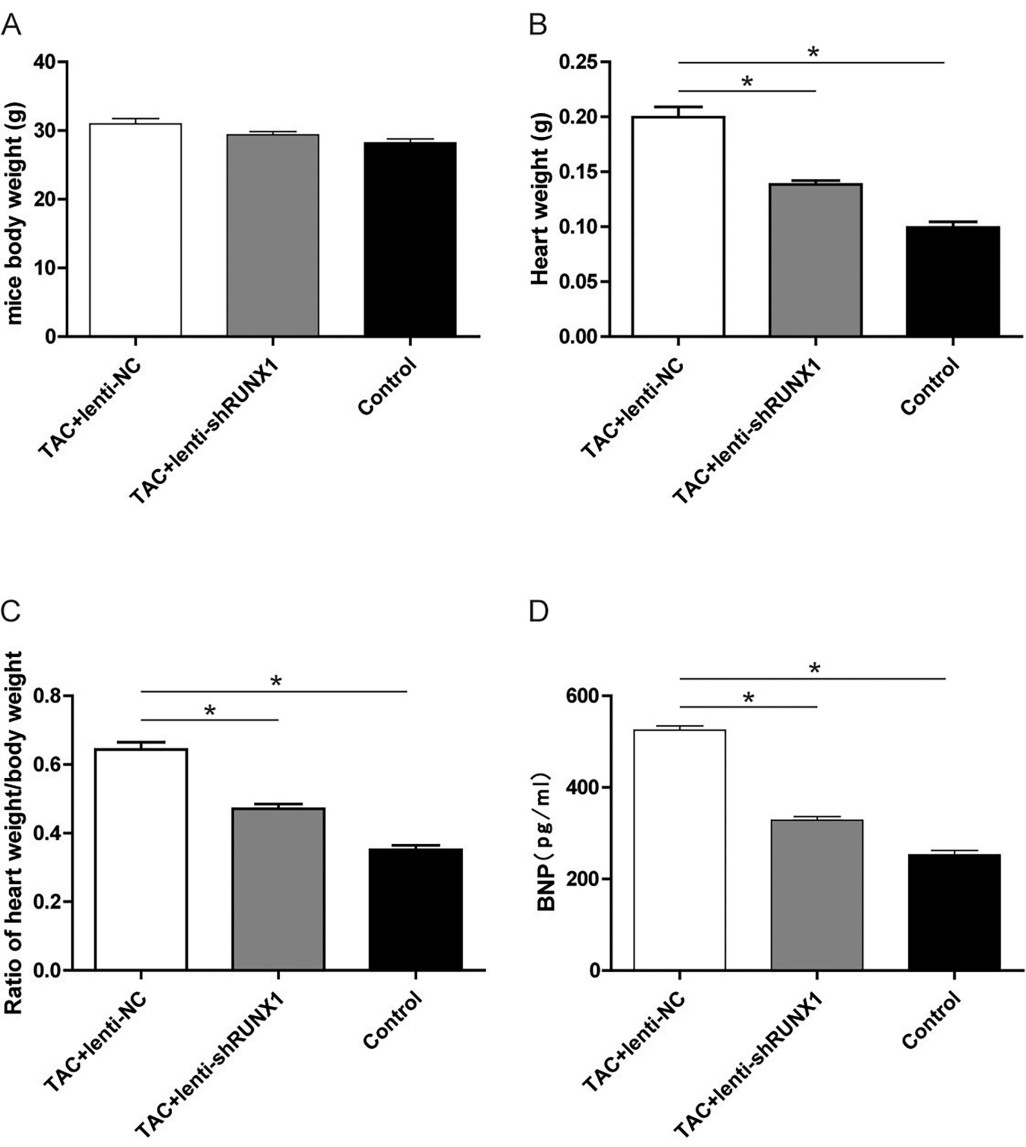

**Figure 5 Detection of mice body weight.** Detection of mice body weight in control, TAC+lenti-NC, TAC+lenti-shRUNX1 groups (A). Detection of heart weight in different groups (B); Ratio of heart weight/body weight in different groups (C); Detection of serum BNP level in different groups (D). Asterisks (*) indicate a significant difference.

## Inhibition of RUNX1 reduces cell apoptosis after TAC induced heart failure

Decrease of myocardial cells caused by apoptosis is one of the important etiologies in heart failure pathophysiological process. We detected cell apoptosis in myocardial tissues using TUNEL staining. The results showed that the myocardial cell apoptosis in TAC+lenti-NC group was significantly higher than that in TAC+ lenti-shRUNX1 group and control group (Fig. 4). This result suggested that the inhibition of RUNX1 could lead to reduction of cell apoptosis in TAC induced model. Besides, this result partly explained why RUNX1 could regulate heart failure process in animal experiments.

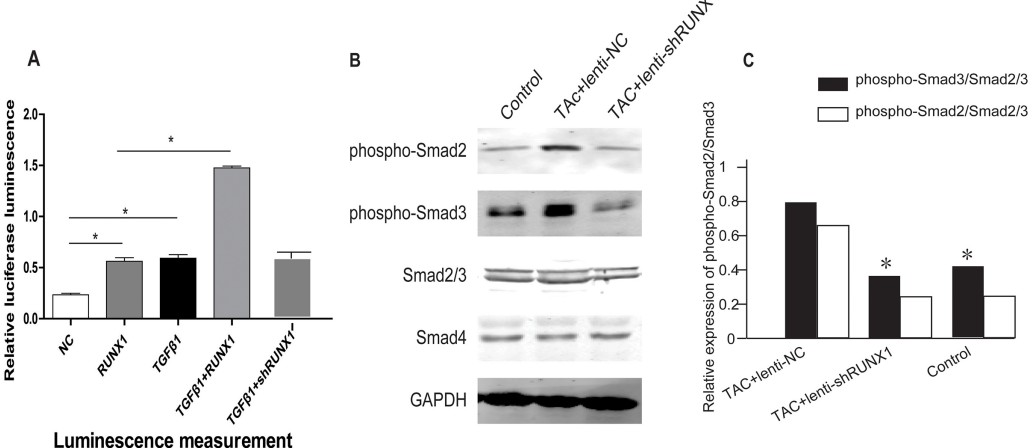

**Figure 6 RUNX1 can activate TGF-β signaling.** (A) Luciferase reporter assay showing that RUNX1 can activate TGF-β signaling in HL-1 cells. TGF-β1 and RUNX1 had cooperative effects in regulating TGF-β signaling. (B) Western blot assay showed that phosphorylation changes of TGF-β downstream signal molecules including Smad2/Smad3 in heart tissues of different groups. Asterisks (*) indicate a significant difference.   

## Inhibition of RUNX1 lead to changes of heart weight/body weight ratio and BNP levels in TAC mouse

Furthermore, we recorded the body weight and heart weight of mice in different groups and compared the heart weight/body weight ratio among them. The results revealed that heart weight/body weight ratio in TAC+lenti-NC group was significantly higher than that in TAC+lenti-shRUNX1 group (Figs. 5A–5C). These macroscopical morphological changes in the heart of TAC mouse indicated that RUNX1 may play an important role in regulating cardiac remodeling that induced by TAC. Moreover, we detected plasma BNP levels in different groups and the results showed that BNP levels in TAC+lenti-NC group were much higher than those in TAC+ lenti- shRUNX1 group (Fig. 5D).

## RUNX1 promotes activation of TGF-β/Smads signaling in mouse heart

In order to illustrate the underlying molecular mechanisms of how RUNX1 regulates myocardial fibrosis, we examined whether the canonical Smad3-dependent TGF-β signaling could be activated by RUNX1 through luciferase reporter assay. The results showed that RUNX1 had significantly stimulated Smad transcriptional activity in HL-1 cells (Fig. 6A). Western blotting results further revealed that the inhibition of RUNX1 caused reduction of phosphorylated Smad2/3 levels in mouse myocardial tissues (Fig. 6B). These results suggested that RUNX1 exerted its effects in cardiac remodeling through regulating the canonical TGF-β signaling pathway.

## DISCUSSION

Heart failure is usually referred to systolic heart failure which is characterized by systolic dysfunction and decline of cardiac output (*Slawinski et al., 2022*). The causes of cardiac systolic dysfunction mainly including decrease of myocardial cells, disorder of myocardial energy metabolism, disorder of excitation-contraction coupling (*Slawson, 2022*). Besides,

myocardial cell apoptosis and necrosis are major causes of myocardial cells decrease (*Wu et al., 2022*). Clinical treatments for heart failure mainly include cardiotonic drugs, β-receptor blockers, angiotensin receptor antagonist, diuretic drugs and vasodilator drugs. All the current medical treatments are lack of specificity with limited clinical efficacy. Therefore, it's urgent to explore new molecular target against heart failure and develop it into anti-heart failure drugs.

Cardiac remodeling is considered as basic mechanism of heart failure initiation and development (*Stassen et al., 2022*). Gross morphological features of cardiac remodeling include ventricular wall thickening, cardiac enlargement, heart weight increase (*Lu et al., 2022*). Besides, histological changes of cardiac remodeling include disorder of myocardial cells arrangement, excess accumulation of collagen and related myocardial fibrosis (*Yi et al., 2022*). The cytological changes of cardiac remodeling include single myocardial cell hypertrophy, fibroblast proliferation with increased collagen synthesis and secretion, increased apoptosis of myocardial cells (*Liang et al., 2022*). Therefore, it is important to find new molecular target that can prevent or ameliorate cardiac remodeling.

The cardiac remodeling can be regulated by several signaling pathways including JAK/STAT, PI3K/AKT, Wnt/β-catenin and TGF-β/Smads et al (*Banks et al., 2022*; *Chen et al., 2022*). Furthermore, it is known that TGF-β/Smads is one of the most important pathways that can regulate cardiac remodeling process. TGF-β1 acts as an essential part in cardiac remodeling and fibrosis and performs its effect through downstream intracellular effectors.

Based on these findings, it is feasible to find new target molecule which can regulate TGF-β/Smads activation during cardiac remodeling process.

Runt-related transcription factor 1 (RUNX1) is considered as a modulator of TGF-β signaling and plays a key role in definitive hematopoiesis (*Lin, 2022*). It is reported that overexpression of RUNX1 can upregulate TGF-β signaling and cause severe blockage in hematopoiesis. The expression of RUNX1 can be augmented by TGF-β in TPO (thrombopoietin)-induced MKs (megakaryocytes) (*Jakobczyk et al., 2022*). As a protein coding gene, diseases associated with RUNX1 include platelet disorder, myeloid malignancy and leukemia (*Masuda et al., 2022*). Studies about RUNX1 related organ fibrosis also emerged in recent years. It is reported that RUNX1 can promote TGF-β-induced renal fibrosis through PI3K subunit p110δ (*Lin, 2022*). The previous studies revealed that RUNX members modulate the transcription of their target genes through recognizing the core binding sequence 5′-TGTGGT-3′, within their regulatory regions *via* runt domain (*Sakurai et al., 2022*; *Zhang, Xia & Lin, 2022*; *Zhong et al., 2022*). In the cardiovascular research field, we searched the published microarray gene expression profile of myocardial infarction (GSE46395) and found that RUNX1 expression was upregulated in myocardial infarction.

The studies above strongly suggested that RUNX1 was not only closely related to TGF-β/Smads signaling but also played an important role in regulating organ fibrosis (*Bao & Guo, 2022*; *Jakobczyk et al., 2022*; *Li & Jia, 2022*).

Herein, the previous studies inspired and encouraged us to perform this study and hypothesize RUNX1 as candidate regulator in cardiac remodeling. Interestingly, we found that the inhibition of RUNX1 could significantly reduce myocardial cell apoptosis and

alleviate cardiac fibrosis induced by TAC in this study. Moreover, our results revealed that the inhibition of RUNX1 could also improve cardiac function in heart failure mouse induced by TAC. These creative results suggested that RUNX1 could be identified as a potential therapeutic target for heart failure treatment.

We innovatively studied the impact of RUNX1 on TGF-β/Smads signaling in myocardial cells and tissues. The results showed that RUNX1 could promote activation of TGF-β/Smads signaling both in myocardial cells and tissues. These results can partly explain how RUNX1 regulates myocardial fibrosis and cardiac remodeling. It is known that TGF-β can induce myocardial cells apoptosis and cell composition changes in myocardial tissue. Therefore, we can conclude from this study that RUNX1 can regulate myocardial cell apoptosis *via* TGF-β/Smads signaling. However, myocardial fibrosis and related cardiac remodeling can be regulated by many signal pathways other than TGF-β/Smads signaling. In addition, it has been revealed that RUNX1 can regulate several signal pathways including NF-κB, Wnt/β-catenin and so on. The Wnt/β-catenin signaling also play an important role in regulating cardiac remodeling. Thus, the mechanisms of how RUNX1 regulates cardiac remodeling need to be further investigated in the future.

In conclusion, the findings in the present study illustrate that the inhibition of RUNX1 can protect against TAC-induced cardiac remodeling and related heart failure. The inhibition of RUNX1 can also improve the cardiac function in TAC-induced mouse. RUNX1 performs its effect in cardiac remodeling mainly through activating TGF-β/Smads signaling and RUNX1/TGF-β/Smads axis could provide new insights in further understanding for mechanisms of heart failure. However, lack of RUNX1 expression and regulation mechanisms was one of the limitations of this study. This study provides RUNX1 as a new potential target for heart failure therapy and new insights for us to understand molecular mechanisms of heart failure initiation.

### Funding

This study was supported by the Natural Science Foundation of Shandong Province (Grant No. ZR2020MH013). The funders had no role in study design, data collection and analysis, decision to publish, or preparation of the manuscript.

### Grant Disclosures

The following grant information was disclosed by the authors:
Natural Science Foundation of Shandong Province: ZR2020MH013.

### Competing Interests

The authors declare that they have no competing interests.

### Author Contributions

- Peng Qi conceived and designed the experiments, performed the experiments, analyzed the data, prepared figures and/or tables, authored or reviewed drafts of the article, and approved the final draft.

- Qian Zhai performed the experiments, analyzed the data, authored or reviewed drafts of the article, and approved the final draft.
- Xiquan Zhang conceived and designed the experiments, analyzed the data, authored or reviewed drafts of the article, and approved the final draft.

## Animal Ethics

The following information was supplied relating to ethical approvals (*i.e.*, approving body and any reference numbers):

This study was approved by the Research Ethics Committee of Qilu Hospital Shandong University (approval number NSFC No. 2022-119).

## Data Availability

Data is available at NCBI GEO: GSE226314.

## Supplemental Information

Supplemental information for this article can be found online at http://dx.doi.org/10.7717/peerj.16202#supplemental-information.

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
