# Peer review of "RUNX1 facilitates heart failure progression through regulating TGF-β-induced cardiac remodeling"

_PeerJ, doi:10.7717/peerj.16202_

## Round 0.1 · original submission · Major Revisions

Dear Dr. Zhang,
Thank you for your submission. Please address the reviewers' concern and revise the manuscript accordingly.
Best,
Jian Song

Reviewer 1 ·

Basic reporting

The language competence, fluency and appropriateness meet publishing requirements. The literature references were added correctly in this manuscript followed the author's guidelines. The article structure was strictly designed and all related figures, table and raw data were provided detailedly in this manuscript.

Experimental design

This original primary research was well designed and consistent with the aim and scope
of this Journal. This research provided well defined and meaningful questions. Based on this aim, this research performed rigorous experiments to answer the scientific questions. Besides, these experiments involved in this study followed high technical and ethical standard. The method in this
study were described detailedly and sufficient.

Validity of the findings

This research identified a new potential molecular target for heart failure treatment and explored its implying mechanisms. All underlying data refer to this study were provided by the authors. The study conclusions were well stated by authors in this manuscript.

Additional comments

This paper addresses an interesting and potentially important topic. This paper focused on RUNX1 function in cardiac remodeling regulation and provided a new potential target against heart failure. The animal experiment in this study revealed the promotive role of RUNX1 in mouse heart failure progression and authors illustrated the potential molecular mechanisms of how RUNX1 regulating cardiac remodeling.
My suggestions of the paper are:

1. Animal anesthesia method need to be described detailedly.
2. In the TUNEL method , please describe the DAPI concentration used in nuclei staining and the stain condition.
3. In the Dual-Luciferase reporter assay, please illustrate the concentration of TGF-β used in HL-1 cell treatment.
4. In the western blot assay, please describe the dilutation ratio of these antibodies including Smad2, phospho-Smad2,Smad3,phospho-smad3.
5. For references 11,please provide the volume and No. of this reference.

Reviewer 2 ·

Basic reporting

no comment

Experimental design

no comment

Validity of the findings

no comment

Additional comments

This study used TAC heart failure mouse model to clarify RUNX1 facilitates heart failure progression through regulating TGF-β-induced cardiac remodeling. This manuscript provided a new target RUNX1 of heart failure and heart remodeling. However, this study needed major revisions before being accepted.
My suggestions about this manuscript as follows,
1. In excel file of figure 1 raw data, all hidden sheets needed checking. Deleting irrelevant hidden sheets in excel of figure 1 raw data is necessary.
2. Why did you choose RUNX1 as a target? Did you use public / In house high throughput omics data to screen and validate abnormal expression of RUNX1 during heart failure?
3. The baseline of RUNX1 expression levels in TAC heart failure model needed evaluating.
4. The efficacy of siRUNX1 needed evaluation.
5. RUNX1 is a transcription factor. However, in “Dual-Luciferase Reporter Assay”, why use recombinant RUNX1 protein treated cells rather than express RUNX1 in the cells. Can RUNX1 protein cross plasma membrane?
6. Did RUNX1 and molecules in TGFβ signaling pathway interact with each other ? Immunoprecipitation (IP) experiments should be performed.
7. The HE staining, Masson’s trichrome staining, and immunofluorescence needed quantification and statistical test.
8. The novelty of this study was low. TGF-β had been well studied in heart failure.

Reviewer 3 ·

Basic reporting

This manuscript lacks the novelty of scientific observation as PMID: 29030345 already has confirmed that RUNX1 deficiency in mice leads to protection against cardiac remodeling upon Myocardial infarction.

Experimental design

However, the presented manuscript presents an experimental procedure without sufficient information and remains uninterpretable.
1. No data is shown of RUNX1 expression levels at basal level and downregulation with lentiviral shRNA.
2. There is no mention of time points of data collection in TAC mice experiments.
3. Fig. 3.-No quantification of collagen deposition (e.g. hydroxyproline assay).
4. Fig. 4. Lacks TUNEL quantification and significance analysis.
5. In Fig. 7. TGF-beta luciferase assay in HL-1 cells, immunoblot lanes do not align with each other to be considered for interpretation. Phosphorylated and unphosphorylated smad2/3 lanes are not from the same origin. Additionally, no quantification is provided for phosphorylation.
6. Fig. 6 RUNX-1 knockdown condition should be included to confirm that upregulated TGF-beta signaling with overexpression of RUX1 with TGF beta can be reversed (similar to mice data).
7. As observed in the myocardial infarction disease model, RUNX1 deficiency does not lead to any change in heart weight in contrary to high heart weight in this manuscript. This manuscript failed to recognize this previous work on RUNX-1 in heart remodeling and no explanations are provided for the contrary observation.

Validity of the findings

No comment

---

## Round 0.2 · accepted · Accept

Hereby confirm I that the authors have addressed all of the reviewers' comments and this manuscript is ready for publication.

Reviewer 1 ·

Basic reporting

The authors have addressed all the issues.

Experimental design

The experimental design is well designed.

Validity of the findings

The findings is interesting for the field.

Reviewer 2 ·

Basic reporting

no comment

Experimental design

no comment

Validity of the findings

no comment

Additional comments

Thank you for your revision. You had answered my question